# Atorvastatin Augments Gemcitabine-Mediated Anti-Cancer Effects by Inhibiting Yes-Associated Protein in Human Cholangiocarcinoma Cells

**DOI:** 10.3390/ijms21207588

**Published:** 2020-10-14

**Authors:** Koh Kitagawa, Kei Moriya, Kosuke Kaji, Soichiro Saikawa, Shinya Sato, Norihisa Nishimura, Tadashi Namisaki, Takemi Akahane, Akira Mitoro, Hitoshi Yoshiji

**Affiliations:** Department of Gastroenterology and Hepatology, Nara Medical University, 840 Shijo-cho, Kashihara, Nara 634-8522, Japan; kitagawa@naramed-u.ac.jp (K.K.); kajik@naramed-u.ac.jp (K.K.); saikawa@naramed-u.ac.jp (S.S.); shinyasato@naramed-u.ac.jp (S.S.); nishimura@naramed-u.ac.jp (N.N.); tadashin@naramed-u.ac.jp (T.N.); stakemi@naramed-u.ac.jp (T.A.); mitoroak@naramed-u.ac.jp (A.M.); yoshijih@naramed-u.ac.jp (H.Y.)

**Keywords:** atorvastatin, HMG-CoA reductase, gemcitabine, TEAD transcriptional activation, yes-associated protein, cholangiocarcinoma

## Abstract

Cholangiocarcinoma (CCA) is associated with high mortality rates because of its resistance to conventional gemcitabine-based chemotherapy. Hydroxy-methyl-glutaryl-coenzyme A reductase inhibitors (statins) reportedly exert anti-cancer effects in CCA and lower the risk of CCA; however, the underlying mechanism of these effects remains unclear. The proliferative and oncogenic activities of the transcriptional co-activator Yes-associated protein (YAP) are driven by its association with the TEA domain (TEAD) of transcription factors; thereby, upregulating genes that promote cell growth, inhibit apoptosis, and confer chemoresistance. This study investigated the effects of atorvastatin in combination with gemcitabine on the progression of human CCA associated with YAP oncogenic regulation. Both atorvastatin and gemcitabine concentration-dependently suppressed the proliferation of HuCCT-1 and KKU-M213 human CCA cells. Moreover, both agents induced cellular apoptosis by upregulating the pro-apoptotic marker *BAX* and downregulating the anti-apoptotic markers *MCL1* and *BCL2*. Atorvastatin also significantly decreased the mRNA expression of the TEAD target genes *CTGF*, *CYR61*, *ANKRD1*, and *MFAP5* in both CCA cell lines. A xenograft tumor growth assay indicated that atorvastatin and gemcitabine potently repressed human CCA cell-derived subcutaneous tumor growth by inhibiting YAP nuclear translocation and TEAD transcriptional activation. Notably, the anti-cancer effects of the individual agents were significantly enhanced in combination. These results indicate that gemcitabine plus atorvastatin could serve as a potential novel treatment option for CCA.

## 1. Introduction

Cholangiocarcinoma (CCA) is the second most common subtype of primary liver neoplasm following hepatocellular carcinoma (HCC) [1]. Although the overall incidence of CCA has gradually increased worldwide over the past four decades, the prognosis of patients with CCA remains poor, with a 5-year survival rate of less than 5% [2,3,4]. Surgical resection remains the primary treatment strategy; however, 20–40% of patients are not eligible for surgery because they were diagnosed with advanced disease. In addition, even among suitable patients, 70% experience recurrence, and the 5-year survival rate is 30% [4,5,6]. A phase III randomized controlled trial [7] defined gemcitabine and platinum-based combination chemotherapy as the standard first-line chemotherapy for unresectable or recurrent CCA. However, the median survival of patients treated with this regimen is less than 1 year [7]. Therefore, other approaches are urgently needed to improve the prognosis of patients with unresectable or recurrent CCA.

Several clinical trials [8,9,10,11,12,13] have sought to prolong the life expectancy of patients with CCA by combining gemcitabine with other cytotoxic drugs, targeted therapeutics, or radiotherapy. However, these combination regimens are often associated with severe adverse effects. Thus, novel molecular targeted agents with acceptable adverse event rates for long-term usage are desired.

Yes-associated protein (YAP) primarily functions with transcriptional co-activator with PDZ-binding motif (TAZ) as transcriptional co-activators associated with the TEA domain (TEAD) family of transcription factors [14,15,16,17]. YAP/TAZ-TEAD elicits a gene expression signature relevant to a variety of oncogenic properties, thereby sustaining proliferation, inhibiting apoptosis, maintaining stemness, regulating metabolism, enhancing angiogenesis, suppressing immune responses, and inducing chemoresistance [17,18,19,20,21,22]. As noted in other cancer types, [21,22,23,24,25,26,27,28,29,30,31] the YAP/TAZ-TEAD pathway plays a crucial role in the progression and prognosis of CCA. Several basic studies [29,32] revealed that the upregulation of TEAD target genes in CCA was associated with increased cancer cell growth, xenograft tumor growth, and resistance to treatment. Moreover, accumulated clinical evidence indicates that intranuclear YAP expression is correlated with worse outcomes in patients with CCA [33]. Therefore, inhibition of YAP/TAZ-TEAD signaling represents an attractive and viable strategy for CCA therapy. Indeed, our recent study [32] demonstrated that angiotensin-II (AT-II) stimulated YAP oncogenic activity by dysregulating the Hippo pathway, an upstream repressor of YAP/TAZ, in human CCA cells. Remarkably, the AT-II type 1 receptor blocker losartan suppressed AT-II–stimulated CCA cell proliferation via YAP inactivation and inhibited intratumor angiogenesis in CCA-derived murine xenograft models.

Hydroxymethylglutaryl coenzyme A (HMG-CoA) reductase inhibitors, also known as statins, are typically used in patients with hypercholesterolemia to inhibit serum lipid synthesis [34]. Interestingly, recent epidemiological studies [33,35] reported that statin use might be associated with a reduced risk of biliary tract cancer, and statins can also potentially enhance the chemotherapeutic efficacy of several anti-cancer drugs such as S-1 and gefitinib against CCA. Furthermore, several lines [29,36] of evidence suggest that statins could suppress YAP activation in some cancer cell types. However, the effect of statin in combination with gemcitabine on the progression of CCA and the mechanism underlying the statin-mediated suppression of oncogenic YAP are unknown.

Several varieties of statins are currently available in clinical practice. However, there are three reasons why we selected atorvastatin in this study: (1) The ideal statin should be fat-soluble, because solubility in fat means good permeability into tissue and infiltration into the cytosol of each cell. (2) The ideal statin should not be excreted into bile. Instead, it should be metabolized in the liver by cytochromes because patients with cholangiocarcinoma usually suffer from cholestasis. (3) The ideal statin should have a strong inhibiting effect on the mevalonate pathway as this will translate into a strong inhibition for cancer cell growth.

In this study, we demonstrated the combinatorial effects of atorvastatin and gemcitabine on CCA cell proliferation via YAP inactivation using CCA-derived murine xenograft models. Based on these findings, we believe statins could have roles in clinical CCA treatment.

## 2. Results

### 2.1. Gemcitabine Plus Atorvastatin Efficiently Suppressed Human Cholangiocarcinoma Cell Growth

Initially, we assessed the effects of gemcitabine and atorvastatin on in vitro human CCA cell growth. A cell proliferation assay illustrated that atorvastatin concentration-dependently attenuated the viability of HuCCT-1 cells according to its traditional mechanism (Figure 1A). Strikingly, gemcitabine also concentration-dependently inhibited the proliferative potential of HuCCT-1 cells (Figure 1B). Moreover, combined treatment with atorvastatin and gemcitabine profoundly enhanced the inhibitory effects of the individual agents (Figure 1C). Next, we examined the effects of both drugs on cellular apoptosis in human CCA cells. As shown in Figure 1D, the activity of caspase-3, a critical executioner of apoptosis, was significantly elevated in human CCA cells treated with atorvastatin and/or gemcitabine. In addition, the number of immunopositive cells was significantly larger in the atorvastatin- and the gemcitabine-treated group as compared to the cleaved caspase 3 (Figure 1E,F).The mRNA levels of *BAX*, a pro-apoptotic marker, were mildly elevated by either treatment alone versus the control, although the differences were not statistically significant (Figure 2A). Meanwhile, combination treatment with both drugs significantly upregulated *BAX* expression in HuCCT-1 cells (Figure 2A). By contrast, the mRNA levels of *MCL1* and *BCL2*, two anti-apoptotic markers, were remarkably decreased by the combination treatment (Figure 2B,C). As for the cell cycle marker, cyclin D1 gene expression levels were also significantly decreased by the combination treatment (Figure 2D). The protein levels of pro- and anti-apoptotic markers indicated the compatible results of each gene expression level (Figure 2E,F). To validate these pharmacological actions on CCA growth, we also evaluated the effects of both agents in another CCA line, KKU-M213. In line with the results in HuCCT-1 cells, both atorvastatin and gemcitabine attenuated proliferation and induced apoptosis in KKU-M213 cells (Appendix A).

### 2.2. Gemcitabine Plus Atorvastatin Inhibited YAP/TAZ-TEAD Activation in Human CCA Cells

To elucidate the effects of both agents on YAP/TAZ-TEAD transcriptional activation, we next investigated the expression levels of TEAD target genes including *CTGF*, *CYR61*, *ANKRD1*, and *MFAP5*. Atorvastatin significantly decreased the mRNA expression levels of these TEAD target genes in HuCCT-1 cells (Figure 3A–D).

Interestingly, these genes were also downregulated following treatment with gemcitabine. The inhibitory effects of both agents in combination were more potent than those of either drug alone (Figure 3A–D). Additionally, we confirmed the effects of both agents in KKU-M213 cells (Appendix A).

### 2.3. Gemcitabine Plus Atorvastatin Reduced the Human CCA Cell-Derived Xenograft Tumor Burden

Given the in vitro inhibitory effects of gemcitabine plus atorvastatin on human CCA cell growth, we assessed the effect of both agents on the in vivo growth of xenograft CCA tumors in athymic nude mice. HuCCT-1 cell-derived xenograft tumors grew progressively in the vehicle-treated control mice but more slowly in mice treated with atorvastatin (100 mg/kg/day) or gemcitabine (100 mg/kg/3 days) (Figure 4A,B). After the experiments, the mean tumor volumes and weights were significantly lower in mice treated with either atorvastatin or gemcitabine than in those treated with vehicle (Figure 4C,D). Moreover, combination treatment resulted in enhanced growth suppression compared with the effects of each drug alone (Figure 4C,D). We next quantitatively evaluated intratumor cell viability via immunohistochemistry, which illustrated that atorvastatin or gemcitabine alone potently abrogated Ki67-positive cell proliferation (Figure 4E,F), while simultaneously inducing TdT-mediated dUTP nick end labeling (TUNEL)-positive cell apoptosis in the tumors derived from HuCCT-1 cells (Figure 5A,B). Notably, combination treatment profoundly enhanced the abrogation of intratumor cell proliferation (Figure 4E,G) and induction of cellular apoptosis (Figure 5A,B). Consistent with the results in cultured HuCCT-1 cells, the intratumor mRNA levels of *BAX* were significantly elevated in atorvastatin- or gemcitabine-treated mice compared with those in vehicle-treated mice (Figure 5C). Meanwhile, both *MCL2* and *BCL2* mRNA levels were reduced in mice treated with both drugs (Figure 5D,E). Similar to the in vitro findings, the effects of both agents on xenograft tumor burden were validated using a KKU-M213 cell xenograft model (Appendix A).

### 2.4. Gemcitabine Plus Atorvastatin Suppressed YAP/TAZ-TEAD Activation in Human CCA Cell-Derived Xenograft Tumors

YAP is activated via translocation into the nucleus, in which it can bind to transcriptional factors such as TEAD.14 Thus, we performed immunohistochemical analyses to evaluate the intracellular localization of YAP in xenograft tumors to determine whether atorvastatin- and gemcitabine-mediated antitumor effects involve the regulation of YAP. As shown in Figure 6A, YAP localized to the nucleus and cytoplasm in vehicle-treated mice. Meanwhile, the tumors in mice treated with either atorvastatin or gemcitabine alone displayed limited nuclear localization of YAP. Furthermore, quantitative analysis demonstrated that nuclear YAP accumulation was lower in the tumors of mice treated with both agents than in those of mice treated with either agent alone (Figure 6B). These findings were supported by the significant reduction of the intratumor mRNA expression of TEAD target genes (i.e., *CTGF*, *CYR61*, *ANKRD1*, *MFAP5*) in mice treated with both atorvastatin and gemcitabine (Figure 6C–F).

To show evidence of inhibition in the YAP signaling pathway, we performed the immunocytochemistry of phosphorylated YAP. The number of cells whose cytosol was p-YAP immunopositive was significantly larger in the atorvastatin- and the gemcitabine-treated group (Figure 6G,H).

## 3. Discussion

Statins deplete mevalonic acid, a cholesterol precursor, and inhibit cholesterol synthesis by suppressing HMG-CoA reductase [34] Meanwhile, multiple functions of statins beyond their cholesterol-reducing effects have attracted attention. Several basic reports suggested that statins exerted anti-cancer effects. Zhou et al. [23] claimed that statins could reverse hypoxic resistance and increase the susceptibility of hypoxic human HCC cells to sorafenib. Kamigaki et al. [36] reported that statins induced cell cycle arrest and susceptibility to apoptosis in human CCA cells. Moreover, some large cohort studies [33,35] demonstrated that statin therapy is associated with a lower risk of CCA. These experimental and clinical findings supported our observed effects of statin monotherapy in two CCA cell lines.

Statins have been suggested to have potent inhibitory effects on the oncogenic YAP signaling pathway. Mechanistically, statins facilitate the phosphorylation of YAP and interfere with its nuclear translocation by inhibiting HMG-CoA reductase. The molecular mechanism underlying this effect is based on the HMG-CoA reductase-mediated activation of the mevalonic acid pathway, more particularly that increased mevalonic acid production triggers the activation of Rho GTPase, which reduces phosphorylated YAP and consequently induces nuclear YAP accumulation [37]. The findings of the current study indicate that atorvastatin suppresses CCA growth by inactivating oncogenic YAP. However, the change in the status of mevalonic acid pathway following atorvastatin treatment was not elucidated in the present study; hence, further investigation is necessary to clarify the molecular basis of atorvastatin-induced YAP regulation in CCA cells.

Gemcitabine-based regimens are often used to treat CCA; however, the development of drug resistance is a major problem. The reason for the poor response of CCA to anti-cancer agents is the existence of complex mechanisms of chemoresistance that usually act synergistically to allow cancer cells to escape from the deleterious effects of cytostatic drugs. Among chemoresistance-related genes, YAP has been reported to promote chemoresistance in various types of cancer cells, including head and neck, lung, breast, stomach, colon, pancreas and ovarian cancers [22,23,24,25,26,27,28,29,30,31]. Several lines of experimental evidence [23,24,30] also demonstrated the close relationship of YAP with chemoresistance in HCC. Furthermore, Marti et al. [29] suggested that YAP reduced the chemosensitivity of CCA cells in conjunction with TEAD transcriptional activation. These findings support our results that atorvastatin attenuates chemoresistance to gemcitabine via YAP inactivation in human CCA cells.

The results also demonstrated that gemcitabine unexpectedly inhibited TEAD target genes and the nuclear translocation of YAP. Basically, it is recognized that intracellular YAP/TEAD activation is dependent on the density of cancer cells. Specifically, low cell density leads to YAP activation and cell proliferation, whereas high cell density suppresses YAP activation, thereby inducing cell contact inhibition. Given these mechanisms, chemotherapy-mediated cell death should activate intracellular YAP as a result of decreased cell density, which was contrary to our results. This discrepancy suggests that gemcitabine itself also can inhibit YAP activation. Although the detailed mechanism of gemcitabine-mediated YAP inhibition is obscure, a recent study [38] using pancreatic cancer cells reported that gemcitabine induced apoptosis and autophagy by activating AMP-activated protein kinase (AMPK), which is also a powerful repressor of YAP activation. Based on these findings, we suspect that gemcitabine-mediated YAP inactivation possibly involves AMPK activation.

The difference between the atorvastatin-treated group and controls in the current analysis of our study was somewhat similar to that between the combined group and the gemcitabine-treated group, though synergistic anticancer effects of gemcitabine with pitavastatin on pancreatic cancer cell line have been recently reported [39]. Based on this information, we deduced that the combination of atorvastatin and gemcitabine bore an additive antitumor effect, and this would be attributed to the completely independent antitumor effect between atorvastatin and gemcitabine; the former acts as an inhibitor of mevalonate pathway and the latter works as a substance with a pyrimidine structure. In addition, the current doses of statin and gemcitabine in vitro as well as in vivo were similar to those selected in some recent reports and to be acceptable for clinical applications.

Several limitations were apparent in the present study. First, other functional mechanisms have been suggested to be associated with the statin-mediated suppression of CCA oncogenic properties. For example, Buranrat et al. [40] reported that atorvastatin-induced reactive oxygen species formation and inhibited cell proliferation. Additionally, Yang et al. [41] observed that lovastatin could inhibit cell growth and adhesion by downregulating transforming growth factor-β and focal adhesion kinase, respectively. Second, Seeree et al. [42] claimed that the inhibitory effects of statins on CCA cells were limited under cholesterol-rich conditions. Thus, further investigation is required to elucidate whether atorvastatin efficiently inhibits CCA cell growth in mice with hypercholesterolemia. Third, inhibition of oncogenic YAP activity also alters the tumor microenvironment, including tumor-associated fibrosis, angiogenesis, and immunity [43]. Additional analysis of these multifunctional effects of statins using the present model is required.

In summary, we successfully demonstrated the combinatorial effect of atorvastatin and gemcitabine on CCA cell proliferation via YAP inactivation using CCA-derived murine xenograft models. The combination of statins and cytotoxic drugs in anti-cancer treatment regimens may permit the use of smaller doses of cytotoxic anti-cancer drugs, possibly leading to reduced side effects. Therefore, we believe that statins could have novel roles in clinical CCA treatment.

## 4. Materials and Methods

### 4.1. Compounds and Cell Culture

Atorvastatin and gemcitabine were purchased from Sawai Pharmaceutical Co. Ltd. (Osaka, Japan) and Tokyo Chemical Industry Co. Ltd. (Tokyo, Japan), respectively. HuCCT-1 and KKU-M213 cells were purchased from the Japanese Collection of Research Bioresources Cell Bank (Osaka, Japan). HuCCT-1 cells were cultured in Roswell Park Memorial Institute 1640 medium (Nacalai Tesque, Inc., Kyoto, Japan) supplemented with 10% fetal bovine serum and 1% ampicillin/streptomycin. KKU-M213 cells were cultured in Dulbecco’s modified Eagle’s medium (Thermo Fisher Scientific K.K., Kanagawa, Japan) supplemented with 10% fetal bovine serum and 1% ampicillin/streptomycin.

### 4.2. Cell Proliferation Assay

Cell proliferation was assessed using the water-soluble tetrazolium salt (WST)-1 assay (Takara Bio Inc., Kusatsu, Japan) according to the manufacturer’s protocol. Briefly, 3 × 10^3^–5 × 10^3^ CCA cells per well were seeded into 96-well plates. After cell adherence, the cells were treated with different concentrations of atorvastatin (0–100 μM) and gemcitabine (0–1.0 μM) for 24 h. Subsequently, cells were incubated with WST-1 for 4 h, and then the absorbance (420–480 nm) was measured using a microplate reader. The anti-proliferative effect was determined as the ratio of the value in the treatment groups to that in the vehicle group. To evaluate the effects of atorvastatin and gemcitabine on cell proliferation, cells were also treated with 20 μM atorvastatin and/or 1 × 10^−2^ μM gemcitabine.

### 4.3. Measurement of Caspase-3 Activity

To assess in vitro cellular apoptosis, caspase activity was detected using a caspase-3 kit (ab39401) purchased from Abcam (Cambridge, UK) according to the manufacturer’s protocol. Caspase-3 recognizes the sequences DEVD and LEHD, and cleavage of the labeled substrate p-nitroaniline results in the emission of light, which was quantified using a spectrophotometer at 405 nm.

### 4.4. Human CCA Cell Xenografts

Six-week-old male athymic nude mice (BALB/cSlc-nu/nu) were obtained from Japan SLC, Inc. (Shizuoka, Japan) and used for the in vivo studies. Mice were housed in stainless steel mesh cages under controlled conditions (temperature: 23 ± 3 °C, relative humidity: 50 ± 20%, 10–15 air changes/h, illumination: 12 h/day). The animals were provided with tap water *ad libitum* during the entire experimental period. For tumor inoculation, 1 × 10^6^ cells were suspended in 200 μL of medium containing Matrigel at a high concentration (Corning, Tewksbury, MA, USA; 1:1) and injected subcutaneously into the bilateral flanks of each mouse. Each tumor was measured using a caliper, and its size was calculated using the following formula: *volume* = (*width*)^2^ * *length*/2. Seven days after inoculation, the tumor-bearing mice were treated with oral atorvastatin at a dose of 100 mg/kg/day, intraperitoneal gemcitabine at a dose of 100 mg/kg twice a week, or both. Mice in the vehicle group were administered an equivalent volume of saline solution (*n* = 5) as a control group. All mice were sacrificed 35 days after administration. The tumors were collected, and their sizes and weights were recorded. All animal procedures complied with the recommendations of the Guide for Care and Use of Laboratory Animals (National Research Council of Japan), and the study was approved on July 17, 2018 by the animal facility committee of Nara Medical University (authorization number: #12347).

### 4.5. RNA Extraction and Quantitative Real-Time PCR

Total RNA was extracted from frozen liver and terminal ileum tissue samples as well as from whole cell lysates using a RNeasy mini kit (Qiagen, Tokyo, Japan) per the manufacturer’s instructions. Next, total RNA was reverse-transcribed into cDNA using a High-Capacity RNA-to-cDNA kit (Applied Biosystems, Foster City, CA, USA) according to the manufacturer’s instructions. Reverse transcription-quantitative PCR with gene-specific primer pairs (Table 1) was performed using a Step One real-time PCR kit (Applied Biosystems). Relative gene expression levels were determined using glyceraldehyde-3-phosphate dehydrogenase as the internal control. The relative level of target mRNA per cycle was determined by applying a threshold cycle to the standard curve. All reactions were performed using a 1:10 dilution of cDNA, whereas mRNA expression levels were estimated using the 2^−ΔΔCt^ method and presented as relative values to each control group.

### 4.6. Immunohistochemical Analyses

Subcutaneous tumors were fixed overnight at 4 °C in 10% formalin and embedded in paraffin. Subsequently, tumors were sliced into 5-µm paraffin sections. Immunohistochemical analyses were performed using paraffin-embedded tumor sections subjected to heat-mediated antigen retrieval using sodium citrate buffer (pH 6.0) for 20 min. Rabbit anti-Ki67 (Abcam, Cambridge, England; 1:100 dilution) and rabbit anti-YAP (Cell Signaling Technology, Danvers, MA, USA; 1:400 dilution) were used as the primary antibodies, and staining was performed according to the suppliers’ instructions. A goat anti-rabbit biotinylated secondary antibody was used to detect the primary antibodies and visualized using a horseradish peroxidase-conjugated ABC system (Vector Laboratories, Burlingame, CA, USA). DAB was used as the chromogen. TUNEL-positive CCA cells were detected using an In-Situ Cell Death Detection Kit (Sigma-Aldrich, Inc., St. Louis, MO, USA) as recommended for tissue sections by the supplier. The sections were counterstained with hematoxylin. Specimens for histology and immunohistochemistry were observed under an optical microscope (BX53; OLYMPUS, Tokyo, Japan) equipped with a digital microscope camera (DP20-5; OLYMPUS). NIH Image J software (http://imagej.nih.gov/ij/) was used for quantitative analyses. All quantitative analyses were performed using five fields per section in high-power fields at 400-fold magnification.

### 4.7. Immunocytochemical Analyses

Cultured cells on the chamber slides were fixed for 15 min at room temperature in 4% paraformaldehyde phosphate buffer solution (Wako, Tokyo Japan) and permeabilized in 0.5% Triton-X/PBS. Five percent goat serum/PBS with 0.1% Tween 20 (BMS, Tokyo, Japan) was used as blocking buffer. Immunocytochemical analyses were performed using rabbit anti-cleaved caspase 3 antibody (#9661, Cell Signaling Technology, Danvers, MA, USA; 1:400 dilution) as the primary antibody and a goat anti-rabbit biotinylated secondary antibody was used to detect the primary antibody.

### 4.8. Immunofluorescence

Fixation, permeabilization, and blocking methods were done as mentioned. Immunofluorescence was performed using rabbit anti-phospho-YAP (Ser127) antibody (Cell Signaling Technology, Danvers, MA, USA; 1:50 dilution) as the primary antibody and a donkey anti-rabbit IgG Alexa Fluor 488 as a secondary antibody (Thermo Fisher Scientific, Waltham, MA, USA; 1:500 dilution). Nuclear was counter stained by DAPI fuluoromount-G (Southern Biotech, Birmingham, AL, USA) according to the manufacturer’s instructions.

### 4.9. Enzyme-Linked Immunosorbent Assay (Elisa) for Pro- and Anti-Apoptotic Markers

The amounts of *BAX* and *BCL2* in in vitro samples were determined using an Elisa kit (Abcam, Cambridge, England) according to the manufacturer’s instructions.

### 4.10. Statistical Analysis

Data were analyzed using either Student’s *t*-test or one-way analysis of variance followed by Bonferroni’s test for multiple comparisons, as appropriate. All tests were two-tailed, and a probability value of less than 0.05 was considered statistically significant.

## Figures and Tables

**Figure 1 ijms-21-07588-f001:**
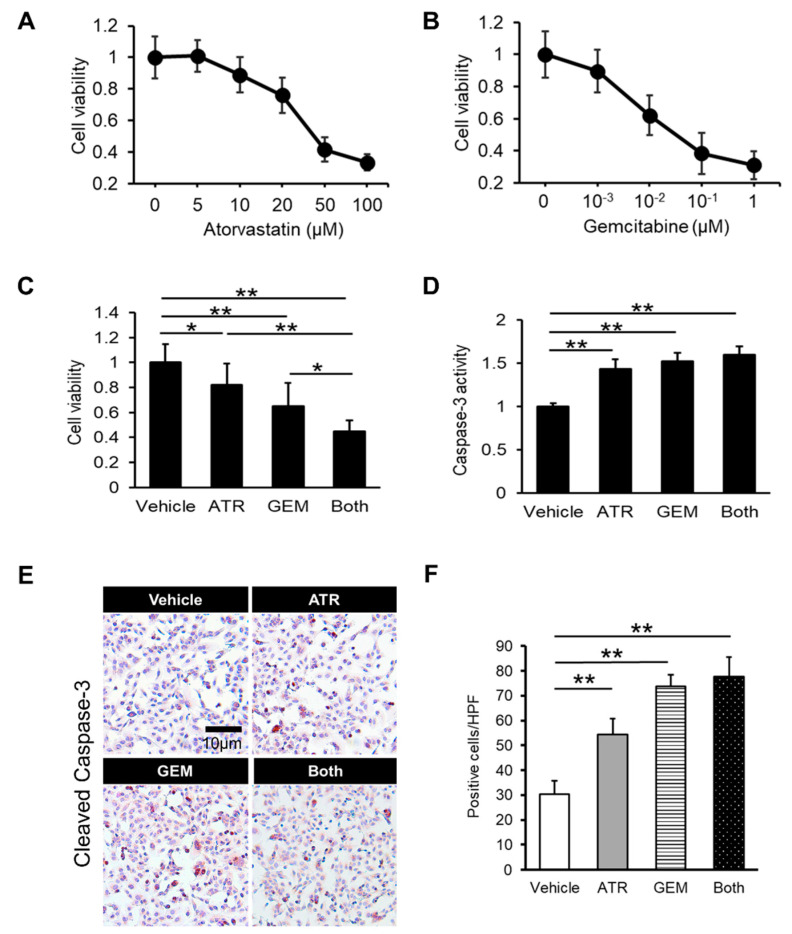
Effects of atorvastatin and gemcitabine on in vitro HuCCT-1 cell growth. (**A**,**B**) Concentration-dependent effects of atorvastatin (0–100 μM) (**A**) and gemcitabine (0–1 μM) (**B**) on the growth of HuCCT-1 cells. (**C**) Combined effect of atorvastatin (20 μM) and gemcitabine (1 × 10^−2^ μM) on HuCCT-1 cell growth. (**D**) Combined effect of atorvastatin (20 μM) and gemcitabine (1 × 10^−2^ μM) on caspase-3 activity in HuCCT-1 cells. (**E**) Immunocytochemistry of cleaved caspase 3 (CC3). (**F**) Quantification of CC3-immunopositive cells. ATR, treatment with 20 μM atorvastatin. GEM, treatment with 1 × 10^−2^ μM gemcitabine. Both, combined treatment with 20 μM atorvastatin and 1 × 10^−2^ μM gemcitabine. Data are presented as the mean ± SD (*n* = 10). * *p* < 0.05; ** *p* < 0.01.

**Figure 2 ijms-21-07588-f002:**
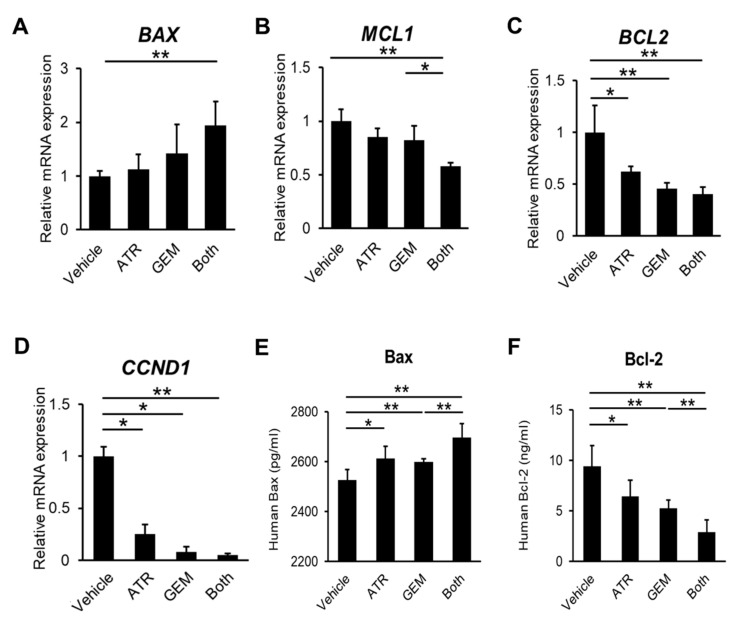
Effects of atorvastatin and gemcitabine on in vitro HuCCT-1 apoptosis. (**A**–**D**) Relative mRNA levels of *BAX* (**A**), *MCL1* (**B**), *BCL2* (**C**), and *CCND1* (**D**) in HuCCT-1 cells. The mRNA expression levels were measured using quantitative real-time PCR, and *glyceraldehyde-3-phosphate dehydrogenase* was used as an internal control. Quantitative values are indicated as ratios of the values of the vehicle-treated group. ATR, treatment with 20 μM atorvastatin. GEM, treatment with 1 × 10^−2^ μM gemcitabine. Both, combined treatment with 20 μM atorvastatin and 1 × 10^−2^ μM gemcitabine. (**E**,**F**) Protein expression levels of *BAX* and *BCL2*. Data are presented as the mean ± SD (*n* = 10). * *p* < 0.05; ** *p* < 0.01.

**Figure 3 ijms-21-07588-f003:**
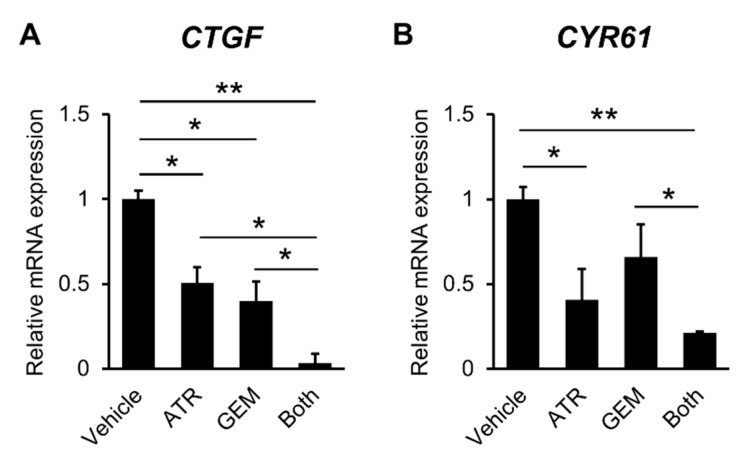
Effects of atorvastatin and gemcitabine on the expression of Yes-associated protein (YAP)/TEA domain (TEAD) target genes in cultured HuCCT-1 cells. (**A**–**D**) Relative mRNA levels of the YAP/TEAD target genes (**A**) *CTGF*, (**B**) *CYR61*, (**C***) ANKRD1*, and (**D**) *MFAP5* in cultured HuCCT-1 cells. The mRNA expression levels were measured using quantitative real-time PCR, and glyceraldehyde-3-phosphate dehydrogenase was used as an internal control. Quantitative values are indicated as ratios relative to the values of vehicle-treated cells. ATR, treatment with 20 μM atorvastatin. GEM, treatment with 1 × 10^−2^ gemcitabine. Both, combined treatment with 20 μM atorvastatin and 1 × 10^−2^ gemcitabine. Data are presented as the mean ± SD (*n* = 10). * *p* < 0.05; ** *p* < 0.01.

**Figure 4 ijms-21-07588-f004:**
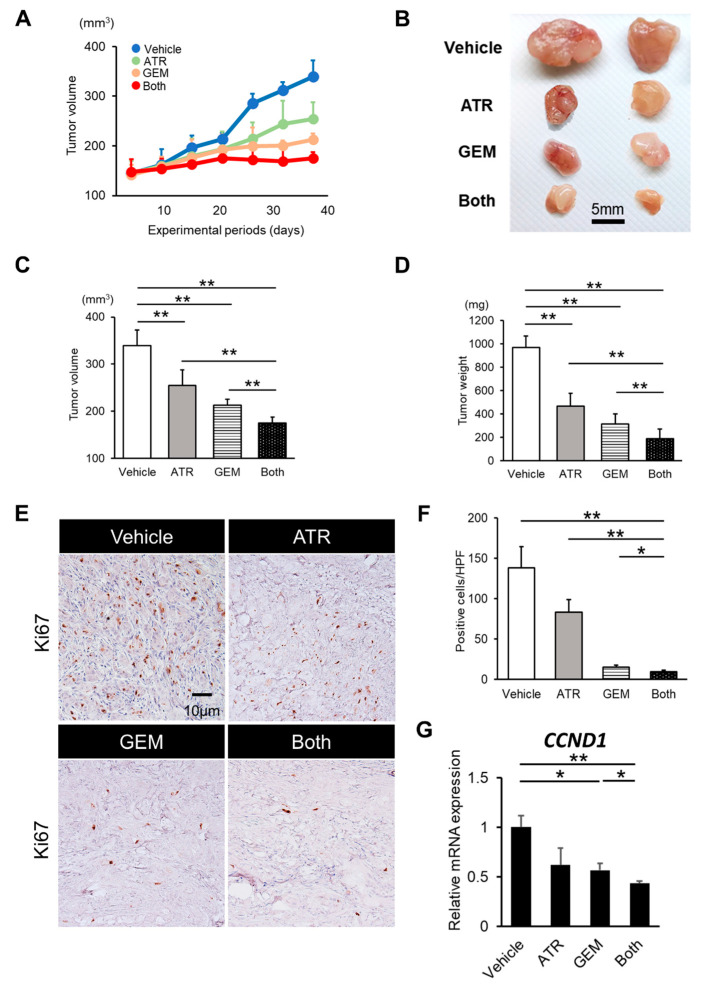
Gemcitabine plus atorvastatin suppressed HuCCT-1 cell-derived tumor growth and inhibited tumor cell proliferation in xenograft model. (**A**) HuCCT-1 cell-derived xenograft tumors grew progressively in vehicle-treated control mice, but they grew more slowly in mice treated with either atorvastatin (100 mg/kg/day) or gemcitabine (100 mg/kg/3 days). (**B**) Representative photographs of resected subcutaneous tumors. (**C**,**D**) The mean tumor volumes (**C**) and weights (**D**) of mice treated with atorvastatin, gemcitabine, or both. ATR, atorvastatin-treated mice. GEM, gemcitabine-treated mice. Both, mice treated with both atorvastatin and gemcitabine. Data are presented as the mean ± SD (*n* = 10). * *p* < 0.05; ** *p* < 0.01. (**E**) Representative pictures of HuCCT-1 cell-grafted subcutaneous tumors stained for Ki67. (**F**) Quantification of Ki67-positive cells. The number of immunopositive cells in high-power fields was counted. (**G**) Relative mRNA levels of *CCND1* in HuCCT-1 cell-derived xenograft tumors. The mRNA expression levels were measured using quantitative real-time PCR, and *glyceraldehyde-3-phosphate dehydrogenase* was used as an internal control. Quantitative values are indicated as ratios of the values of the vehicle-treated group.

**Figure 5 ijms-21-07588-f005:**
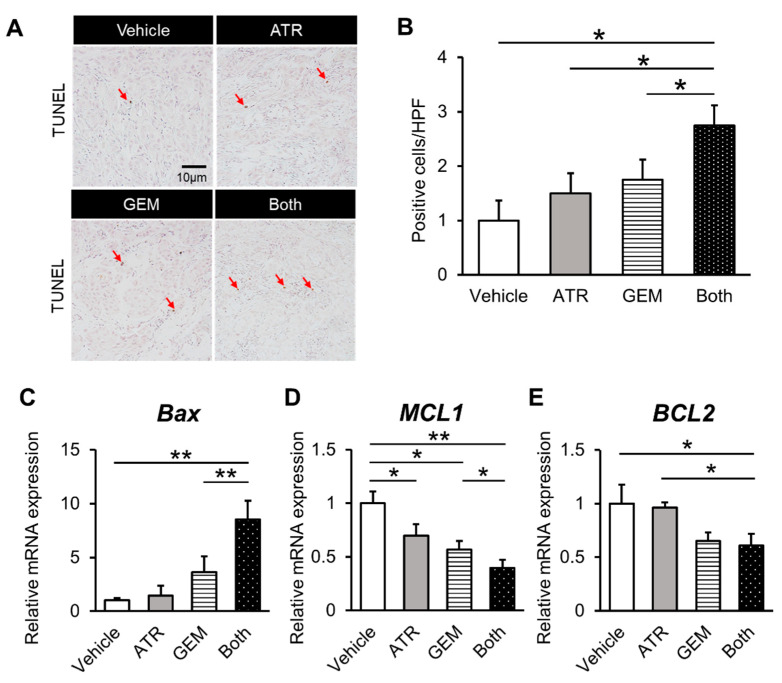
Gemcitabine plus atorvastatin promoted apoptosis in HuCCT-1 cell-derived xenograft tumors. (**A**) Representative pictures of HuCCT-1 cell-grafted subcutaneous tumors stained for TdT-mediated dUTP nick end labeling (TUNEL). Red arrows indicate immunopositive cells. (**B**) Quantification of TUNEL-positive cells. The number of immunopositive cells in high-power fields was counted. (**C**–**E**) The intratumor mRNA levels of *BAX* (**C**), MCL2 (**D**), and *BCL2* (**E**). The mRNA expression levels were measured using quantitative real-time PCR, and glyceraldehyde-3-phosphate dehydrogenase was used as an internal control. Quantitative values are indicated as ratios relative to the values of the vehicle-treated group. ATR, atorvastatin-treated mice. GEM. gemcitabine-treated mice. Both, mice treated with both atorvastatin and gemcitabine. Data are presented as the mean ± SD (*n* = 10). * *p* < 0.05; ** *p* < 0.01.

**Figure 6 ijms-21-07588-f006:**
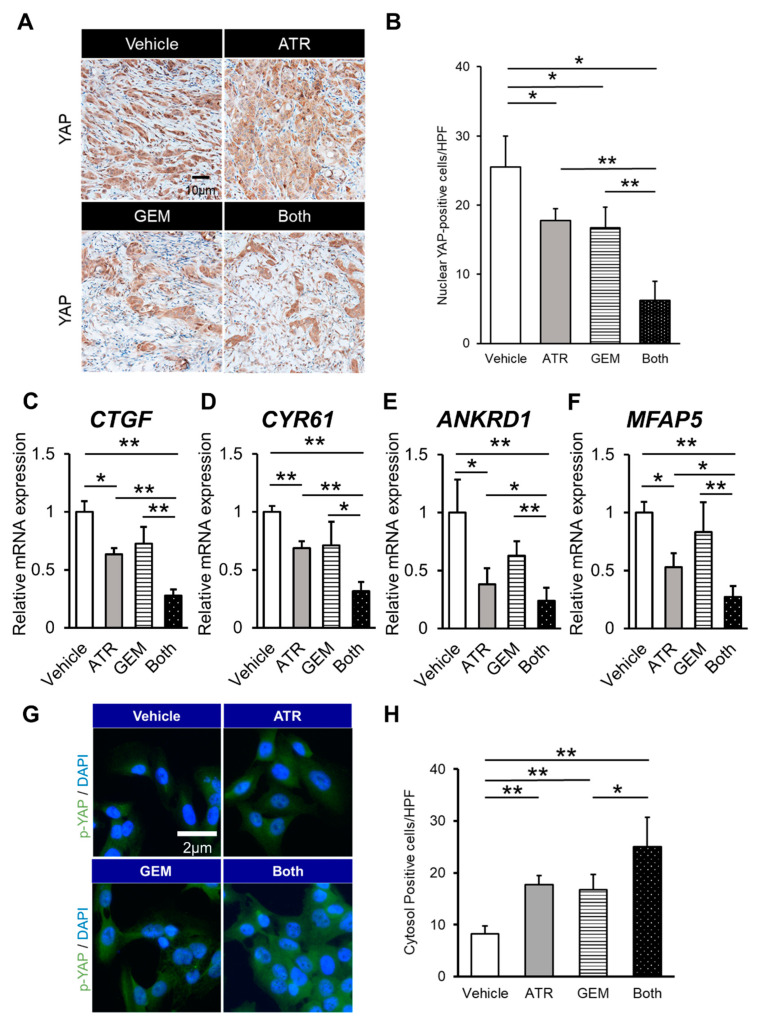
Gemcitabine plus atorvastatin interfered with Yes-associated protein (YAP) nuclear shuttling in HuCCT-1 cell-derived xenograft tumors. (**A**) Representative pictures of HuCCT-1 cell-grafted subcutaneous tumors stained for YAP. (**B**) Quantification of nuclear YAP-positive cells. The number of immunopositive cells in high-power fields was counted. (**C**–**F**) Relative mRNA expression of *CTGF*, *CYR61*, *ANKRD1*, and *MFAP5* in HuCCT-1 cell-grafted subcutaneous tumor tissues. The mRNA expression levels were measured using quantitative real-time PCR, and glyceraldehyde-3-phosphate dehydrogenase was used as an internal control. Quantitative values are indicated as ratios relative to the values of the vehicle-treated group. (**G**) Representative pictures of HuCCT-1 cells immunofluorescently stained for phospholylated YAP. (**H**) The number of immunopositive cells in their cytosols in high-power fields was counted. ATR, atorvastatin-treated mice. GEM, gemcitabine-treated mice. Both, mice treated with both atorvastatin and gemcitabine. Data are presented as the mean ± SD (*n* = 10). * *p* < 0.05; ** *p* < 0.01.

**Table 1 ijms-21-07588-t001:** List of primers for quantitative real-time PCR.

Gene	Sense (5′–3′)	Antisense (5′–3′)
*CTGF*	TGCTTTGAACGATCAGACAA	CTTGTGGCAAGTGAATTTCC
*CYR61*	AAGAAACCCGGATTTGTGAG	GCTGCATTTCTTGCCCTTT
*ANKRD1*	GCCTACGTTTCTGAAGGCTG	GTGGATTCAAGCATATCACGGAA
*MFAP5*	GTGACTCAAGCGACTCCAGAA	AGTCATCTGTGGAAGGTGCAAT
*BAX*	TCTGACGGCAACTTCAACTG	GGAGGAAGTCCAATGTCCAG
*MCL1*	GAGGGCGACTTTTGGCTAC	GTACCCGTCCAGCTCCTCTT
*BCL2*	CATGTGTGTGGAGAGCGTCAA	GCCGGTTCAGGTACTCAGTCA
*CCND1*	CCCTCGGTGTCCTACTTCAA	CTTAGAGGCCACGAACATGC

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
