# Peer review of "Atorvastatin Augments Gemcitabine-Mediated Anti-Cancer Effects by Inhibiting Yes-Associated Protein in Human Cholangiocarcinoma Cells"

_ijms, 2020, doi:10.3390/ijms21207588_

Round 1

Reviewer 1 Report

General comment

In general, this paper is nicely construed, well-written and clear. Experiments are easy to understand and follow. The introduction provides the right amount of background information with one exception that is why atorvastatin was selected for this study and not other statins.

The nature of the presented work is correlative/observational, not mechanistic.

I have a number of concerns about the strength of what is presented. Therefore, what the authors are claiming is only partially supported by the provided data.

I did not check for plagiarism.

Specific comments

Results/Figure 1. There is confusion between the written result section relative to Figure 1. In fact, Fig1A and Fig.1B appear to be swapped. Consequently, also Supplementary Fig1 might be wrong.

There is a bit of confusion regarding the effects of the two drugs. Effects both on proliferation and cell death are presented.  Similarly, the xenograft experiments also show both PCNA and TUNEL assay. I would recommend providing a FACS cell cycle analysis and an AnnexinV staining. This would help to better understand the effects of the two compounds alone or in combination.

Along these lines, it would strengthen the paper if the authors showed also protein expression of pro-apoptotic and anti-apoptotic molecules in addition to the mRNA. It is notorious that in some instances protein expression does not correlate with mRNA.

The authors should also clarify two important points: 1-establish whether the effects are additive (as it appears) or synergistic. 2- Are the doses used in vitro (this is also a request for the doses used in mice) achievable in patients?

Since in the title the authors claim that the observed effects are mediated by inhibition of YAP, the authors should provide some direct evidence of this inhibition and not just discuss it. It would be desirable to perform immunoblots to show the phosphorylation status of YAP and WB on nuclear and cytoplasmic fractions.

Finally, there is a recurrent concern regarding the number of samples and biological replicates that were performed to obtain the illustrated results.

In the figure legends is reported that n = 10. But, the xenograft experiment reports that groups were of 5 mice. So, please clarify for each experiment (both in vitro and in vivo) what is the n for each group (controls vs treated). Please, also specify if those are coming from independent experiments or are from the same one experiment.

Author Response

Reviewer 1

General comment

  1. In general, this paper is nicely construed, well-written and clear. Experiments are easy to understand and follow. The introduction provides the right amount of background information with one exception that is why atorvastatin was selected for this study and not other statins. The nature of the presented work is correlative/observational, not mechanistic. I have a number of concerns about the strength of what is presented. Therefore, what the authors are claiming is only partially supported by the provided data. I did not check for plagiarism.

Thank you for this valuable suggestion. Actually, several studies have already been reported regarding the antitumor effects of statins. The reasons why we rationally selected atorvastatin for this study are the following: 1) The ideal statin should be fat-soluble, because solubility in fat means good permeability into tissue and infiltration into the cytosol of each cell. 2) The ideal statin should not be excreted into bile. Instead, it should be metabolized in the liver by cytochromes because patients with cholangiocarcinoma usually suffer from cholestasis. 3) The ideal statin should have a strong inhibiting effect on the mevalonate pathway as this will translate into a strong inhibition for cancer cell growth.

With these points in mind, atorvastatin was the most feasible drug, hence we decided to use this statin in the current study.

We added the following sentence in the introduction section: Several varieties of statins that are actually available in the current clinical practice. However, there are three reasons why we selected atorvastatin in this study: 1) The ideal statin should be fat-soluble, because solubility in fat means good permeability into tissue and infiltration into the cytosol of each cell. 2) The ideal statin should not be excreted into bile. Instead, it should be metabolized in the liver by cytochromes because patients with cholangiocarcinoma usually suffer from cholestasis. 3) The ideal statin should have a strong inhibiting effect on the mevalonate pathway as this will translate into a strong inhibition for cancer cell growth.

Specific comments

  1. Results/Figure 1. There is confusion between the written result section relative to Figure 1. In fact, Fig1A and Fig.1B appear to be swapped. Consequently, also Supplementary Fig1 might be wrong.

Thank you very much for this comment. We interchanged gemcitabine and atorvastatin in the result section (Please see the line 136, 138, 140, and 162).

  1. There is a bit of confusion regarding the effects of the two drugs. Effects both on proliferation and cell death are presented. Similarly, the xenograft experiments also show both PCNA and TUNEL assay. I would recommend providing a FACS cell cycle analysis and an AnnexinV staining. This would help to better understand the effects of the two compounds alone or in combination.

Thank you for this very important comment. Following your suggestion, we individually separated the results of proliferation and cell death (shown in the new Figure 4 and the new Figure 5). In addition, and according to the reviewer’s comments, we added the result of cyclin D1 gene expression both in vitro and in vivo as a cell cycle analysis (shown in the new Figure 2D and the new Figure 4G). Furthermore, we also added the results of immunocytochemistry for the cleaved caspase 3 (shown in the new Figure 1E) and counted the number of immunopositive cells in each group (shown in the new Figure 1F).

We added the following sentences in the results section: “In addition, the number of immunopositive cells was significantly larger in the atorvastatin- and the gemcitabine-treated group as compared to the cleaved caspase 3 (Figure 1E and 1F).” and “As for the cell cycle marker, cyclin D1 gene expression levels were also significantly decreased by the combination treatment (Figure 2D).”

  1. Along these lines, it would strengthen the paper if the authors showed also protein expression of pro-apoptotic and anti-apoptotic molecules in addition to the mRNA. It is notorious that in some instances protein expression does not correlate with mRNA.

We appreciate your comment. According to your suggestion, we demonstrated the protein expression levels of pro-apoptotic and anti-apoptotic factors (shown in the new Figure 2E and 2F).

We added the following sentence in the results section: “The protein levels of pro- and anti-apoptotic markers indicated the compatible results of each gene expression level (Figure 2E and 2F).”

  1. The authors should also clarify two important points: 1-establish whether the effects are additive (as it appears) or synergistic. 2- Are the doses used in vitro (this is also a request for the doses used in mice) achievable in patients?

1) Thank you for important comments. Although synergistic anticancer effects of gemcitabine with pitavastatin on the pancreatic cancer cell line have recently been reported (Cancer Manag Res. 2020; 12: 4645-65), the differences between the atorvastatin-treated and control groups in the current study were somewhat similar to that between the combined group and the gemcitabine-treated group. Based on this information, we deduced that the combination of atorvastatin and gemcitabine bore an additive antitumor effect, and this would be attributed to the completely independent antitumor effect between atorvastatin and gemcitabine; the former acts as an inhibitor of mevalonate pathway while the latter works as a substance with a pyrimidine structure. 2) The dose setting in vitro was based on the results of the preliminary experiments which are shown in the new Figures. 1A and 1B. These concentrations (Atorvastatin: 20 μM, Gemcitabine: 0.01 μM) are also similar to previous reports. (Please refer to the article: Cancer Manag Res. 2020; 12: 4645-65, Biol Pharm Bull. 2017; 40: 1247-54, Int J Mol Sci. 2014; 15: 8106-21.) The dose setting in vivo was based on the basic experimental data on those developmental stages that were available to the public from pharmaceutical companies. Ideal doses of test drugs in rodents have been generally thought as 3–4 times higher than in humans. Given that the effective dose of gemcitabine in preclinical in vivo studies in mice has been 80–160 mg/kg while the usual dose in humans in clinical practice has been 25–30 mg/kg (1000 mg/m2), the current gemcitabine dosage (100 mg/kg) in mice could be applied to clinical practice. With regard to atorvastatin, in light of the established safe dose of 70–125 mg/kg based on a 52-week daily dosing study in rodents and the maximum low-density lipoprotein cholesterol suppression effect reached at approximately 30 mg/kg, the current atorvastatin dosage (100 mg/kg) in mice is also considered to be clinically applicable. Based on these facts, our dose settings in vitro and in vivo are available for use in clinical applications.

We added the following sentence in the discussion section: “The difference between the atorvastatin-treated group and controls in the current analysis of our study was somewhat similar to that between the combined group and the gemcitabine-treated group, though synergistic anticancer effects of gemcitabine with pitavastatin on pancreatic cancer cell line have been recently reported (Cancer Manag Res. 2020; 12: 4645-65). Based on this information, we deduced that the combination of atorvastatin and gemcitabine bore an additive antitumor effect, and this would be attributed to the completely independent antitumor effect between atorvastatin and gemcitabine; the former acts as an inhibitor of mevalonate pathway and the latter works as a substance with a pyrimidine structure. In addition, the current doses of statin and gemcitabine in vitro as well as in vivo were similar to those selected in some recent reports and to be acceptable for clinical applications.”

  1. Since in the title the authors claim that the observed effects are mediated by inhibition of YAP, the authors should provide some direct evidence of this inhibition and not just discuss it. It would be desirable to perform immunoblots to show the phosphorylation status of YAP and WB on nuclear and cytoplasmic fractions.

We appreciate your thoughtful comment. To meet your requirements, we performed the immunocytochemistry (shown in the new Figure 6G) and counted the number of phosphorylated YAP immunopositive cells in its cytosol (shown in the new Figure 6H).

We added the following sentence in the result section: To show evidence of inhibition in the YAP signaling pathway, we performed the immunocytochemistry of phosphorylated YAP. The number of cells whose cytosol was p-YAP immunopositive was significantly larger in the atorvastatin- and the gemcitabine-treated group (Figure 6G and 6H).

  1. Finally, there is a recurrent concern regarding the number of samples and biological replicates that were performed to obtain the illustrated results. In the figure legends is reported that n = 10. But, the xenograft experiment reports that groups were of 5 mice. So, please clarify for each experiment (both in vitro and in vivo) what is the n for each group (controls vs treated). Please, also specify if those are coming from independent experiments or are from the same one experiment.

Thank you very much for having reviewed our article. In the in vivo assay, we assigned five mice in each group in both experiments with HuCCT-1 and KKU-M213 related xenograft model. Each mouse was inoculated with tumor cells (HuCCT-1 or KKU-M213) in both flanks as indicated in the materials and methods. Subsequently, we obtained a total of ten tumors from five recipient mice which were used for the related analyses. Concerning the in vitro assays, we set each ten cell-culture dish in every group and used for the related analysis.

We also added the following sentences in the materials and methods section:

Immunocytochemical analyses

Cultured cells on the chamber slides were fixed for 15 minutes at room temperature in 4% paraformaldehyde phosphate buffer solution (Wako, Tokyo Japan) and permeabilized in 0.5% Triton-X/PBS. Five percent goat serum/PBS with 0.1% Tween 20 (BMS, Tokyo, Japan) was used as blocking buffer. Immunocytochemical analyses were performed using rabbit anti-cleaved caspase 3 antibody (#9661, Cell Signaling Technology, Danvers, MA, USA; 1:400 dilution) as the primary antibody and a goat anti-rabbit biotinylated secondary antibody was used to detect the primary antibody.

Immunofluorescence

Fixation, permeabilization, and blocking methods were done as mentioned. Immunofluorescence was performed using rabbit anti-phospho-YAP (Ser127) antibody (Cell Signaling Technology, Danvers, MA, USA; 1:50 dilution) as the primary antibody and a donkey anti-rabbit IgG Alexa Fluor 488 as a secondary antibody (Thermo Fisher Scientific, MA, USA; 1:500 dilution). Nuclear was counter stained by DAPI fuluoromount-G (Southern Biotech, Birmingham, USA) according to the manufacturer’s instructions.

Enzyme-linked immunosorbent assay (Elisa) for pro- and anti-apoptotic markers

The amounts of BAX and BCL2 in in vitro samples were determined using an Elisa kit (Abcam, Cambridge, England) according to the manufacturer’s instructions.

Reviewer 2 Report

The publication "Atorvastatin augments gemcitabine-mediated anti-cancer effects by inhibiting Yes-associated protein in human cholangiocarcinoma cells" is a very interesting scientific study. Research results have a potential clinical aspect. All of the publications, i.e., abstact, introduction, results and discussion, are synthesized and understandable to the reader. The results are documented with clear figures. The "Materials and Methods" section is described accordingly. It is very important that the  Authors use two experimental models, i.e. in vitro and in vivo.

The publication is based on the results of other researchers, mostly published in the last 10 years. Citing recent scientific reports is another positive feature of this publication.

Author Response

Reviewer 2

The publication "Atorvastatin augments gemcitabine-mediated anti-cancer effects by inhibiting Yes-associated protein in human cholangiocarcinoma cells" is a very interesting scientific study. Research results have a potential clinical aspect. All of the publications, i.e., abstact, introduction, results and discussion, are synthesized and understandable to the reader. The results are documented with clear figures. The "Materials and Methods" section is described accordingly. It is very important that the Authors use two experimental models, i.e. in vitro and in vivo. The publication is based on the results of other researchers, mostly published in the last 10 years. Citing recent scientific reports is another positive feature of this publication.

Thank you very much for this warm comment and having reviewed our article.

We also added the following sentences in the materials and methods section:

Immunocytochemical analyses

Cultured cells on the chamber slides were fixed for 15 minutes at room temperature in 4% paraformaldehyde phosphate buffer solution (Wako, Tokyo Japan) and permeabilized in 0.5% Triton-X/PBS. Five percent goat serum/PBS with 0.1% Tween 20 (BMS, Tokyo, Japan) was used as blocking buffer. Immunocytochemical analyses were performed using rabbit anti-cleaved caspase 3 antibody (#9661, Cell Signaling Technology, Danvers, MA, USA; 1:400 dilution) as the primary antibody and a goat anti-rabbit biotinylated secondary antibody was used to detect the primary antibody.

Immunofluorescence

Fixation, permeabilization, and blocking methods were done as mentioned. Immunofluorescence was performed using rabbit anti-phospho-YAP (Ser127) antibody (Cell Signaling Technology, Danvers, MA, USA; 1:50 dilution) as the primary antibody and a donkey anti-rabbit IgG Alexa Fluor 488 as a secondary antibody (Thermo Fisher Scientific, MA, USA; 1:500 dilution). Nuclear was counter stained by DAPI fuluoromount-G (Southern Biotech, Birmingham, USA) according to the manufacturer’s instructions.

Enzyme-linked immunosorbent assay (Elisa) for pro- and anti-apoptotic markers

The amounts of BAX and BCL2 in in vitro samples were determined using an Elisa kit (Abcam, Cambridge, England) according to the manufacturer’s instructions.

Round 2

Reviewer 1 Report

The authors have adequately addressed my concerns.